# Study Circles as a Possible Arena to Support Self-Care—A Swedish Pilot Study

**DOI:** 10.3390/ijerph21040483

**Published:** 2024-04-15

**Authors:** Birgitta Kerstis, Jorgen Herlofson, Lena Wiklund Gustin

**Affiliations:** 1School of Health, Care and Social Welfare, Mälardalen University, 722 20 Västerås, Sweden; birgitta.kerstis@mdu.se; 2Empatica AB, 752 20 Uppsala, Sweden; j.herlofson@gmail.com; 3Department of Health and Care Sciences, UiT The Arctic University of Norway, Campus Narvik, 8514 Narvik, Norway

**Keywords:** mental health, mixed methods, self-care, self-compassion, study circles

## Abstract

Today, issues related to people’s mental health and well-being have been described as a challenge for society, globally as well as in Sweden. This calls for new approaches to mental health promotion. The aim was to evaluate the adequacy of its content and structure, describing experiences of study circles as a means of supporting participants’ self-care and self-compassion. The overall design is a descriptive QUAL + quan design, where the quantitative and qualitative results are integrated. Five participants participated in a focus group interview, of whom four completed questionnaires. One individual interview was conducted with the study circle leader. Study circles can be an arena for mental health promotion, as learning and sharing of experience contributes to a sense of coherence, as well as self-compassion and a genuine concern for one’s own and others’ well-being, but are not considered an alternative to psychiatric care for those in need of professional services. Study circles can be a possible means to support self-care and thereby promote mental health in the general population and are a valuable contribution to public health. However, in addition to modifications of the content, further research is needed on the qualifications for study circle leaders, as well as the dissemination of study circles.

## 1. Introduction

Today, issues related to people’s mental health and well-being have been described as a challenge for society, globally [1] as well as in Sweden [2]. This has been a problem for decades, but the COVID-19 pandemic further added to this problem [3]. These figures do not include people with psychiatric diagnoses. Rather, they represent people struggling with symptoms such as worries, anxiety, insomnia, and altered mood that negatively affect their well-being and quality of life without fulfilling all the criteria needed for a psychiatric diagnosis according to the DSM-5. Consequently, more people turn to healthcare facilities with problems that are not within the realm of the service providers’ responsibility. It has been argued that the incorporation of “mental ill health” into ordinary language has contributed to the psychiatrization of ordinary life problems [4]. In addition, people who frequently contact healthcare services describe problems in coping with their ordinary lives [5]. Psychological distress is also associated with poor work ability [6]. This is also in line with a person-centered approach to mental health as an experience of life as meaningful and oneself as capable of managing the problems encountered in life [7].

Educational groups that include self-compassion and are organized by the healthcare system and led by professionals can contribute to experiences of stress reduction, well-being, and confidence in managing stressful situations in different clinical contexts outside psychiatric care [8,9]. However, as the World Health Organization (WHO) [10] guidelines put forth that self-care must work as an extension of the health system, it is important to explore new approaches to strengthening people’s self-care skills at the societal level. In this article, we will describe a pilot study where we introduced a study circle (SC) focusing on self-care and self-compassion, as the latter has been suggested to be as important to self-care as compassion is to the care of others [11].

Even though SCs exist in different parts of the world, their status in Sweden as a mass phenomenon has been described as rather unique [12]. Their history dates to the first decades of the 20th century, when they provided an opportunity for people to educate themselves and gain knowledge outside of university [13]. Today, SCs are organized by study organizations, such as the People’s University (Folkuniversitetet) and the Worker’s Educational Association (ABF). Their content can vary from theoretical, more or less academic courses to practical skills training. SCs are partly financed by state subsidies and led by laymen with knowledge and skills in a specific subject [12]. Hence, they could be a possible format for disseminating knowledge on self-care and mental health outside healthcare organizations. Previous research is sparse, but SCs have been presented as easily administered and cost-effective health-promoting interventions [14].

Orem [15] highlights the importance of supporting patients in upholding their self-care needs. However, self-care cannot be reduced to knowledge and skills. For example, nurses do have sufficient knowledge about the importance of self-care, yet they often fail to execute self-care in their own lives [16]. To succeed, self-care needs to involve both genuine concern and compassion for oneself [17].

Self-compassion includes three aspects: (1) the ability to be consciously present and available to oneself and in touch with one’s feelings (mindfulness), (2) understanding that to be human is to be vulnerable and that no one is perfect (common humanity), and (3) the ability to be kind, show concern, and not be judgmental towards oneself (self-kindness) [18]. Self-compassion is also associated with less psychopathology and stress, as well as increased resilience and self-care [19,20]. Another concept that is significant for understanding individuals’ self-care and mental health is Sense Of Coherence (SOC). The concept originates from a salutogenic approach and suggests that individuals with a strong SOC perceive life events as comprehensible, manageable, and meaningful, ultimately leading to better health outcomes and overall well-being [21]. All three aspects are important to people’s experience of health and can also be seen as an indication that a person has the ability to cope with the challenges they encounter in life.

### Context and Aim

The point of departure for the SC was a therapeutic group program on a cognitive, relational basis (CRGP) that was originally developed for the treatment of patients on long-term sick leave in connection with work-related depression [22]. The CRGP was modified for preventive purposes among people with stress-related health problems and shorter periods of sick leave [23]. The CRGP was modified further to be used as a SC.

The SC consisted of an introductory session and six group sessions of 90 min which took place 1–2 weeks apart. The structure and content are described in Table 1. In addition, the participants received written material to support their learning and further reflections.

To evaluate the adequacy of its content and structure, we designed a pilot study aiming to describe the experiences of the SC as a means for supporting participants’ self-care and self-compassion. Thus, the pilot study sought answers to how the participants experienced the SC as a means of developing self-care and whether the SC contributed to an increased SOC and self-compassion. The first question also provided an opportunity to identify areas for further development of the SC.

## 2. Methods

### 2.1. Design

In this study, the design was informed by Dattilio, Edwards, and Fishman’s [24] recommendation to use case studies within a mixed methods approach, as well as Clark’s [25] impetus on accounting for the interplay between theories and how this informs practice when developing new treatments. The case is our pilot SC and the theoretical point of departure the person-centered approach to mental health described above, as well as an understanding of self-care and self-compassion as contributing to experiences of mental health. The overall design is a QUAL + quan mixed methods design [26], where descriptive statistics are integrated with the presentation of the qualitative findings.

### 2.2. Participants and Recruitment

People were invited to the SC in conjunction with an open lecture about mental health and self-care in the municipality’s library. Five participants volunteered and participated in the SC. The inclusion criterion was self-perceived stress in daily life. Thus, there were no diagnostic criteria for inclusion. The exclusion criteria included ongoing sick leave or treatment for stress or any other form of mental ill health. However, one participant did not complete the questionnaires. The participants were women aged between 42 and 59 years old, all of whom were employed and had a structured social life, despite grappling with various challenges in their personal and professional lives. The study circle leader (SCL) was a licensed cognitive therapist with many years of experience both in individual and group psychotherapy, including the CRGP.

### 2.3. Data Collection

The qualitative data were collected after the last session. The third author conducted a focus group dialogue with the participants and a semi-structured interview with the SCL. In the focus group dialogue, the participants were encouraged to recall and jointly reflect on their experiences of the SC and its impact on their self-care and experiences of well-being. The interview with the SCL focused on his experiences of the possibilities and challenges associated with leading a SC. The focus group dialogue and the interview were transcribed verbatim.

In addition, the participants answered questionnaires on three occasions, at baseline, after the last session, and at a three-month follow-up, including the Self-Compassion Scale (SCS) [27] and Sense of Coherence (SOC-29) [28]. The SCS is a 26-item self-report questionnaire that reflects the three dimensions of self-compassion and their counterparts [27]. SOC-29 has a salutogenic perspective on health and refers to comprehensibility, manageability, and meaningfulness in life [28] and has a strong relationship with mental health and self-care ability [29].

### 2.4. Analyses

The qualitative data were analyzed using Graneheim and Lundman’s [30] inductive approach to qualitative content analysis with cooperation between the first and the third author. The analysis commenced by seeking units of meaning within the transcripts based on their relevance to the study’s aim and research questions, excluding text only when the participants departed from the subject. These units were then condensed and coded. The codes were then compared to identify similarities and differences, leading to the emergence of nine subcategories and three categories, which were ultimately synthesized into a theme. While categories and sub-categories describe *how* a phenomenon (i.e., SCs) is experienced, the theme linking them together focuses on describing *what* is experienced at an abstract level [30]. The process is exemplified in Table 2.

Concerning the quantitative analyses, the SOC and SCS scales are presented with mean scores and standard deviations. To measure the effect size, Cohen’s d was used to present the differences in terms of the standard deviation units in relation to the SC.

### 2.5. Ethical Considerations

The study was approved by the Swedish Ethical Review Authority (Dnr: 2021-04928) and in accordance with the Declaration of Helsinki. Information regarding the study’s aim, the confidentiality of the data handling, and the study’s voluntariness were provided verbally and in writing, and the participants gave their written informed consent. To provide a safe environment for sharing experiences, issues relating to confidentiality between group members were jointly reflected on in the informational meeting.

## 3. Results

The presentation of the findings is structured around the categories and sub-categories that arose from the qualitative analysis, ending up with a theme describing the SC as an *arena to create changes* (Table 3).

An overview of the quantitative results is presented in Table 4 and Table 5. In line with the QUAL + quan mixed methods design, these results are reflected in the presentation of the qualitative findings below. In summary, there was a significant change between the total SOC 1 and SOC 2, *t* (3) = −2.08, *p* = 0.025, *d* = 4.16, but no significant differences between the subscales. There was no statistically significant change between the total SCS scores. Among the subscales, there were significant changes between Self-kindness 1 and Self-kindness 2, *t* (3) = −4.90, *p* = 0.016, *d* = 2.45, and Self-judgement 1 and Self-judgement 2, *t* (3) = 8.33, *p* = 0.004, *d* = 4.17.

In the following text, the sub-headings refer to categories, while the sub-categories are integrated into the text and written in *italic*. To adjust for grammar, the naming of them is slightly modified in the text. Quotations have been utilized to elucidate the content of the subcategories. To enhance transparency, these have been labeled with GM and SCL to identify group members and the study circle leader, respectively. The majority of the quotations have been selected from the group members to amplify their voices.

### 3.1. Sharing Experiences with Others

The SC provided an opportunity to share experiences with others on equal terms, *to be mutually engaged in narration*, and thereby receive and give support and compassion.

*“To be able to tell and get compassion from others and so on. When memories fade, you will remember things (like this sharing) that are connected to emotions better.”* (GM)

This was also acknowledged by the SCL, who put forth participants’ engagement and willingness to jointly reflect on their experiences in relation to the SC themes. Sharing experiences in an SC also enabled participants to *normalize emotional struggles.* This was discussed as being less stigmatizing than a therapeutic group session within the context of (mental) health care. Normalizing emotional struggles could also be related to the comprehensibility subscale in the SOC.

Being in a situation where one was involved in joint exploration rather than treatment made a difference and contributed to a sense of *being equal with others*. This was also in line with the SCL’s experience.

*“There is a difference between patients and people who are interested and attending an SC. [---] What they wanted to talk to each other about had an impact and it was much more relational and self-reflective work [than with patients].”* (SCL)

Hence, sharing experiences with others on equal ground contributed to a sense of not being the only one experiencing challenges in life. This corresponds to the results from the SCS and experiences of increased common humanity and decreased isolation.

### 3.2. Developing New Understandings and Competencies

The theoretical material, i.e., the texts provided to read and reflect on between sessions, as well as self-assigned homework, was perceived as adding nuances to the reflections made during and between sessions. This contributed to increased self-awareness, as bodily, emotional, cognitive, existential, relational, and behavioral *experiences became integrated with knowledge.* This awareness made participants more conscious of their own reactions and thus also of possible ways to manage them.

*“So, this is what really matters, to create this sensitivity for when things suddenly change, when the weather changes inside me. Yes, here comes the new weather. What happened? Can I discover (more) or return this way? There are things that are … well, directly applicable.”* (GM)

This could also be understood in terms of being mindful, in contrast to over-identification (SCS), and contributing to *increased self-care competencies.*

*“Self-care can be several different things. How I speak to myself can be one thing, or how I feel can be another. But it can also be keeping routines and managing commitments [---] but also dealing with conflicts.”* (GM)

The SCL also noted that over time, the group participants became more articulate and developed their ability to share emotions with each other. An example is a situation where a participant described a challenging situation for her and how others could provide support.

*“It was a very profiled description of her situation, and in parts quite easy to take in and empathize with.”* (SCL)

Sharing emotions with others, and also being able to set boundaries, thereby emerged as a central relational competence. This could be interpreted as the participant developing new competencies that enabled her to become less self-judgmental and exhibit more self-kindness (SCS). Additionally, it could be seen as an enhanced ability to manage the challenges of daily life, even though the increase in manageability (SOC) was minor. Developing new competencies was also challenging, and the participants sometimes failed/avoided to assign themselves personal homework between sessions. Hence, they highlighted the need for more structured “homework” between sessions as a complement to the written materials. This was described as a means *to stay committed* between sessions and not just read through the material and think, “This was interesting” (GM). The SCL also noted a need to develop general assignments that could give support and directions to the homework but still be flexible enough to be adjusted to the participants’ individual needs.

### 3.3. Navigating New Grounds

While the previous categories are related to the content of the SC, this category departs in addressing the participants’ and the SCL’s reflections on what needs to be considered in future development based on their experiences from the group. The participants as well as the SCL highlighted the need *to balance between structure and freedom*. Finding this balance was perceived as a challenge, especially during the initial meeting. A participant said, “The first time, you waste a lot of energy on setting up in some way”, and the SCL noted that initially, “participants had a bit of difficulty grasping the format, and probably expected me to lecture and educate them, but we soon found our way”. The participants emphasized the importance of the SLC being clear about his role from the beginning.

*“Once you realize that this person will be just as curious about the topics as others in the group, then I think it can work really well, because then I believe the group will adapt to those conditions.”* (GM)

The participants also valued when the group leader assumed a more guiding role during the (few) instances where a participant dominated the conversation excessively, as such behavior from others could be disruptive. Hence, the SCL should lead the way but also give space for participants to elaborate on their reflections, and if necessary, regulate people who tend to dominate and take care of emotional responses that might arise. The SCL also pointed out that the less experienced a SCL is, the higher the need is for structure. There is also a need *to allocate the appropriate time*, both to implement more practical exercises in sessions and to commit to and follow up on homework to create a personalized plan based on the insights provided during the SC.

*“Then you could leave for the last time and have a little plan for yourself that ‘yes, but this is important to me, to take better care of myself, I would have to prioritize this and this and then I need to work a little more on this and that’.”* (GM)

The SC was perceived as a valuable alternative to therapeutic interventions, building on trust in peoples’ capability to address challenges in life as fellow human beings (common humanity, SCS). Hence, the participants also described an urge *to spread the word.*

*“It could be like this wonderful guerrilla activity. You just spread it like this and then people sit and talk on their own and (are able to) take care of it.”* (GM)

Hence, engaging in an SC can be described as a meaningful (SOC) activity if people are willing to invest time in it. The overarching theme, linking the three categories to each other, highlights the SC as an arena for change, as the sharing of experiences with others, as well as the content of the SC, contributed to new understandings and competencies. It was also evident that the fact that this arena existed outside the healthcare system was perceived as an advantage, not least because the dialogues took place on equal terms within the context of the SC. Not surprisingly, both the participants and the SCL described the SC as an opportunity to create change in a wider context.

## 4. Discussion

This pilot study indicates that SCs can be a possible arena for mental health promotion, as learning and sharing of experience contributes to a SOC, as well as self-compassion and a genuine concern for one’s own and others’ well-being. These factors have been associated with people’s psychological well-being [18]. This could partly be understood as related to the content and its focus on mindfulness, as well as different aspects of psychological functioning. However, as described by Kumpusalo and Pitkajarvi [31], there is a relationship between perceived health, self-care activities, and social support. Thus, as the SC is also an opportunity to bring people together, it could also be the human encounters during the SC that contribute to the findings. Self-care activities and social support exist in the context of relational competence, which is considered a crucial aspect of people’s well-being. Relational competence is defined as the collection of qualities that enable individuals to interact effectively with one another [32]. It is demonstrated when person A’s motivation, knowledge, and conversational skills with person B contribute to both participants’ satisfaction with themselves, others, and the communication process [33]. On the one hand, relational competence may be something that participants already possessed before enrolling, contributing to their perception of the SC as helpful. On the other hand, the SC may have also contributed to the development of their relational competence through interaction with each other, as suggested by the progression in the interpersonal process described in the subcategory “sharing experiences with others”. This remains uncertain, and it is likely that a combination of these factors is at play. Therefore, it may be crucial in future research to specifically examine relational competence.

An interesting finding in the analysis of the quantitative data was that both self-compassion (SCS) and SOC continued to increase three months after the SC. Given that this is based on a small number of individuals, it should not be considered conclusive evidence, but it still raises thoughts about the possibility that SCs could contribute to a learning process that continues after the last session and that the self-care competencies the participants acquire may make a difference to their well-being in the long term. This is worth noting, especially as SOC is often perceived as relatively stable [34]. In relation to the SC’s overall theme, self-care, it is also interesting to observe that the significant differences regarding the SCS lie in the subscales of self-kindness and self-judgment, which increased and decreased, respectively. This indicates that the ability to take care of oneself has been positively influenced. However, we do not know for sure whether and how self-compassion and SOC are related to each other. There are numerous studies demonstrating that both aspects are important to individuals’ well-being, particularly concerning stress [35,36,37,38,39]. A recent study also indicates that both SOC and self-compassion (SCS) are important factors in how well healthcare professionals have coped with severe stress during the COVID-19 pandemic [40]. Furthermore, research suggests that personal intelligence may play a mediating role between self-compassion and SOC in relation to anxiety [41]. However, whether this is the case in our study cannot be determined at this point. Nevertheless, we see here that self-care could also be a possible mediator. However, larger quantitative studies are needed in the future to definitively speak to this issue. The results of this limited pilot study, however, suggest that such studies could contribute valuable knowledge in the future.

Nevertheless, the SC was perceived as beneficial to the participants, which calls for further studies. The findings also highlight that further modifications are needed: to set out more time for each session and add further material to the sessions, including more specific between-session homework. SCs may also benefit work groups of nurses, as they do not always translate their knowledge about self-care into their own lives [16].

### Methodological Considerations

This study was undertaken as a pilot study, accounting for the experiences of the participants and the SCL in one SC. In line with Dattilio et al.’s [24] recommendation, we have used a mixed methods design framed as a case study. This allowed us to follow the dissemination of the SC and to merge and reflect on different aspects related to the process. This closeness, and the fact that authors 2 and 3 were involved both in developing and disseminating the SC, is considered an advantage but also a challenge. As researchers 2 and 3 are familiar with the original CRGP [22] and previous modifications of it [23] and have implemented it in different contexts, comparisons and adjustments were facilitated. On the other hand, such closeness could bias the study. To balance this, the first author was invited to be part of the analysis and the interpretation of the data.

It is also worth reflecting on the significance of the participants getting to know each other and developing a comfortable atmosphere during the SC for the focus group discussion. Our hope is that their familiarity with each other contributed to them collectively expressing their experiences in words. We view it as an advantage that the focus group leader (third author) was not involved in the SC and that the interview with the study circle leader (SCL) was conducted after the focus group. This meant that the focus group leader had only a general understanding of the study circle and its content and could remain open during the dialogue. However, it is important to be aware that the risk of social desirability bias [42] exists, particularly if the group’s cohesion led them to want to appear as “good group participants” in front of the researcher.

A limitation was the low number of participants, which makes it difficult to draw conclusions. However, the aim was not to generalize the results to a broader population but to evaluate the SC as a potential format for this type of intervention. A limitation of the surveys is their reliance on self-reports, as the unreliability of human memory is well documented. However, self-report surveys are widely used and often demonstrate high reliability when respondents trust the confidentiality of the study [43]. Therefore, we assert that this pilot study can serve as a foundation for continuing the project, despite the results not being generalizable yet. In addition, we conclude that SOC and the SCS could be used in follow-ups in future, larger studies with more participants.

Despite this potential weakness, we perceive that the data have been rich and have contributed to our understanding of the opportunities and challenges with study circles. By providing detailed descriptions of the participants, the data collection procedures, and the steps taken in the analysis, we have also ensured the trustworthiness of our qualitative content analysis, aligning with the recommendations of Graneheim and Lundman [30,44].

Another issue is that the pilot SC was led by a licensed psychotherapist. It is possible that the mainly positive outcome is linked to the group leader’s competence. However, involving a highly qualified therapist was a deliberate choice at this early stage when piloting the content and structure of the SC. Nevertheless, for a wider dissemination of SCs, there is a need to establish a system where SCLs could be recruited among people with less formal competence. One possibility is to introduce SCs into primary healthcare as a method for health promotion and thus assign professionals who are not psychotherapists, yet who are familiar with self-care issues, as SC leaders.

## 5. Conclusions

SCs are not considered an alternative to psychiatric care for people in need of professional services. Rather, SCs should be understood as a possible means to support self-care and thereby promote mental health in the general population. Hence, SCs can make a valuable contribution to public health. However, in addition to modifications of the content, further research is needed on the qualifications for SCLs, as well as the dissemination of SCs.

## Figures and Tables

**Table 1 ijerph-21-00483-t001:** Overview of the study circles.

Joint Structure for All Sessions	Session Themes
Mindfulness exercise	Session 1: Caring for myself and things that matter in life
Follow-up, reflections on last session’s theme and experiences of homework (1 session, reflection on being in the SC)	Session 2: My emotions–friends or enemies
Group leader’s introduction of a new theme	Session 3: My thoughts–facts or prejudices
Joint reflection on the theme	Session 4: Competencies that can facilitate my everyday life
Self-assigning of homework	Session 5: Communication and interactions in relationships
Mindfulness exercise	Session 6: Everyday living as the basis for a valued life

**Table 2 ijerph-21-00483-t002:** Example of the process of analysis.

Meaning Unit (Quote)	Condensed Meaning Units	Code	Sub-Categories	Categories
When we talked to each other, when we were allowed to share and someone listened, and it’s very special when the group says, “No, what do you say? Oh, tell me more”. I think that is a very important key	Being allowed to share while others are listening is the key	Being listened to	To be mutually engaged in narration	To share experiences with others
Yeah, but I think it’s good that an occasion would have to be like this that you look at everybody’s experiences and how it all ties together and … for me, communication, and relationships, I think, communication to others, communication to myself and … and relationships, relationships to others, relationships to myself	You look at how experiences are tied together, for example, communications and relationships	Reflecting on experiences	To integrate experiences with knowledge	To develop new understandings and competencies
It is required of a conversation leader … to be able to handle strong emotions, to be able to handle silence, and so on, and how do you get people to talk. But also, what tools can you give if you see that this is a person who might need something else?	The conversation leader must be able to guide the conversation and adjust to people’s needs	The SCL has control and is attentive	To balance between structure and freedom	To navigate new ground

**Table 3 ijerph-21-00483-t003:** Overview of qualitative results.

Sub-Categories	Categories	Theme
To be mutually engaged in narration	Sharing experiences with others	An arena to create changes
To normalize emotional struggles		
To be equal with others		
To integrate experiences with knowledge	Developing new understandings and competencies	
To increase one’s self-care competencies		
To stay committed		
To balance between structure and freedom	Navigating new grounds	
To allocate the appropriate time		
To spread the word		

**Table 4 ijerph-21-00483-t004:** Sense of Coherence, SOC. Subscales and total scores.

	Before SC *n* = 4 Mean (SD)	After SC *n* = 4 Mean (SD)	Three Months after SC *n* = 2 Mean (SD)
Meaningfulness	45.00 (5.60)	47.50 (6.24)	45.50 (0.71)
Comprehensibility	45.50 (8.81)	47.00 (10.65)	52.50 (10.61)
Manageability	51.00 (14.21)	52.50 (11.36)	49.00 (14.14)
SOC Total	141.50 (27.14)	147.00 (25.47)	147.00 (25.46)

SC = Study circle.

**Table 5 ijerph-21-00483-t005:** Self-Compassion Scale (SCS). Subscales and total scores (Self-judgement, Isolation, and Over-identification reversed).

	Before SC *n* = 4 Mean (SD)	After SC *n* = 4 Mean (SD)	Three Months after SC *n* = 2 Mean (SD)
Self-kindness	3.05 (1.04)	3.85 (0.81)	4.30 (0.71)
Self-judgment	11.90 (5.12)	10.10 (4.96)	7.90 (5.23)
Common humanity	4.06 (0.24)	3.94 (0.43)	4.12 (0.53)
Isolation	9.44 (4.58)	8.38 (4.10)	9.75 (4.60)
Mindfulness	3.44 (0.66)	4.12 (0.32)	3.75 (0.35)
Over-identification	11.88 (4.05)	9.25 (2.80)	9.00 (5.30)
SCS total	80.50 (18.66)	92.25 (14.31)	95.00 (21.21)

SC = Study circle.

## Data Availability

Since the study was based on a small sample size from which it could be possible to identify individual participants, the data have not been made publicly available. This is in line with how confidentiality issues and data storage were addressed in the ethics review application.

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
