# Peer review of "Study Circles as a Possible Arena to Support Self-Care—A Swedish Pilot Study"

_ijerph, 2024, doi:10.3390/ijerph21040483_

Round 1
Reviewer 1 Report
Comments and Suggestions for Authors
Thank you for this opportunity to review this work. I enjoyed learning about study circles and the integration of qualitative and quantitative data was compelling. I have a few suggestions to add depth to what is already here:
1. Re: the intro, the background on study circles and why these are a promising modality was great. I also appreciated learning why the authors felt self-compassion was important to measure; however, sense of coherence is not mentioned. As this was a key outcome, including background on sense of coherence and why authors feel this scale might be impacted by the study circle participation would be helpful.
A bit more clarity on the aims of this study would be helpful, as it seems there are several. For example, is the aim of the study a first step to understanding how and why study circles might promote well-being, exploring how self-compassion and sense of coherence as potential mediators of change? How is the self-care term related to sense of coherence (or is it)? In addition, was an aim to understand acceptability of study circles, and areas for improvement?
2. In the methods section, I really liked table 2. Authors might specify the way categories and sub-categories were created-- inductively from the transcripts themselves, or deductively, by examining the transcripts for comments related to self-compassion or sense of coherence.
3. In the results section, consider leaving out p values for the quantitative data; given the very small sample size, using 'significant' may be misleading.
In addition, the format of the results where mixed quant/qual data were explored was confusing. Were there participant quotes used? If yes, which comments were from the leader versus the participants? Can you provide more examples or explanation of how participants developed new competencies?
4. In the discussion, it would be enhanced by authors interpretation of the changes in self-compassion and sense of coherence at 3 months. These are interesting trends worth commenting on, and may be also found in other literature.
It also seems there could be more discussion of self-compassion and sense of coherence as mediators to improved well-being-- if this is what authors are implying? How does the format facilitate this? I got the sense, for example, that a big factor identified by participants as beneficial was in community with peers-- listening to, recognizing commonalities with, and relaying stories to peers rather than a hierarchical one-way structure of knowledge passing from "teacher" and "student"-- this seems essential and can be elaborated on/contextualized with the literature.
Reviewer 2 Report
Comments and Suggestions for Authors
Thank you very much for submitting the paper. I enjoyed reading it and reviewing your manuscript.
The paper presents an evaluation of a swedish pilot study on study circles as an beneficial addition to health-care in order to improve well-being public health. Doing so it adresses an important recent topic shifting the perspective to ways to promote mental health.
The research design is promosing - however participant numbers are comperably small. The explanatory power, especially of the quantitative part, should be reflected more cirtically in chapter 4.1.
Line 94/95: Please provide further information on the Mixed-Method design. What does descriptive refer to in this sense?
Line 99: more information on the participants would be promising if available. what kind of struggles lead them to search for help in the SC?
Chapter 3: i dont see how and where qualitative and quantitative data are integrated.
Table 3 "Theme: An area to create changes" - did just one theme emerge? What is the added value to include a special column for that?
Chapter 3.1 to 3.3:
How do you distinguish between direct quote, your interpretaton and results? This section is partly hard to read.
Furthermore i wonder if the participants didn´t mention further, more ciritcal, things to be improved or questioning the value of SC in general? If not - still you should reflect on the interview atmosphere. Did the group discussion lead to social desirability?
Line 239-244: where was this statement taken from? I couldn´t find it in the sections refering to the SC leader. Would it have potential implications for the quality of the SC?
